# SEED: Towards More Accurate Semantic Evaluation For Visual Brain Decoding

**Juhyeon Park**[1*]**, Peter Yongho Kim**[2*]**, Jiook Cha**[1,3]**, Shinjae Yoo**[4]**, Taesup Moon**[1,2,5†]
[1]IPAI, Seoul National University, [2]ECE, Seoul National University,
[3]Psychology, Seoul National University, [4]Brookhaven National Lab,
[5]ASRI / INMC / AIIS, Seoul National University
{parkjh9229, peterkim98, tsmoon}@snu.ac.kr

## Abstract

We present SEED (**Se**mantic **E**valuation for Visual Brain **D**ecoding), a novel metric for evaluating the semantic decoding performance of visual brain decoding models. It integrates three complementary metrics, each capturing a different aspect of semantic similarity between images inspired by neuroscientific findings. Using carefully crowd-sourced human evaluation data, we demonstrate that SEED achieves the highest alignment with human evaluation, outperforming other widely used metrics. Through the evaluation of existing visual brain decoding models with SEED, we further reveal that crucial information is often lost in translation, even in the state-of-the-art models that achieve near-perfect scores on existing metrics. This finding highlights the limitations of current evaluation practices and provides guidance for future improvements in decoding models. Finally, to facilitate further research, we open-source the human evaluation data, encouraging the development of more advanced evaluation methods for brain decoding. Our code and the human evaluation data are available at https://github.com/Concarne2/SEED.

## 1 Introduction

Visual brain decoding focuses on reconstructing visual stimuli from brain signals, such as functional magnetic resonance imaging (fMRI), thereby bridging the fields of neuroscience and computer vision. This field of research is pivotal for developing brain-computer interface (BCI) systems (Mai et al., 2024; Du et al., 2022; Saha et al., 2021) and provides key insights into the working mechanisms of complex human perceptual systems (Mai et al., 2024). Reflecting its importance, numerous studies have been dedicated to advancing this domain (Scotti et al., 2023; 2024; Wang et al., 2024a; Huo et al., 2024; Xia et al., 2024a; Wang et al., 2024b; Tian et al., 2025).

With the recent advent of diffusion-based decoding models (Scotti et al., 2023; 2024; Wang et al., 2024a;b; Huo et al., 2024; Tian et al., 2025) that boast a near-perfect performance on

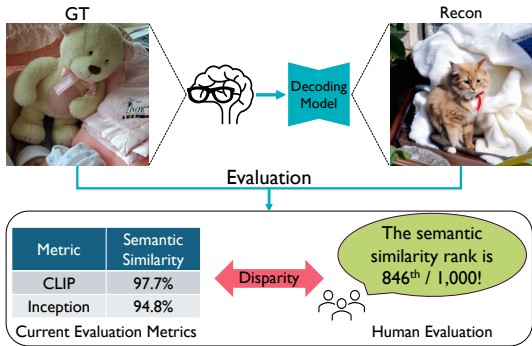

Figure 1: Current evaluation metrics assess the semantic similarity between ground-truth and reconstructions in a way that significantly differs from human evaluation, often giving relatively high scores to reconstructions that are semantically misaligned.

all of the percentage-based evaluation metrics, the endeavor to visually decode brain signals might seem to be nearly solved, with little to no room for improvement for future research. However, upon close inspection, the decoding results, even from the most recent and state-of-the-art models, often

---

*Equal contribution.
†Corresponding author.

fail at reconstructing crucial semantic elements in the original image; *e.g.*, a teddy bear may turn into a cat during the reconstruction process. (See Fig. 1)

As this example suggests, we observed that current evaluation metrics tend to assign relatively high scores to such flawed reconstructions, potentially misleading researchers and obscuring the true limitations of these models. This leads to the following question: *Is the current framework to evaluate visual decoding models aligned with human intuition?* To answer that, we first inspected current evaluation metrics and identified a few limitations: the dependency on the comparison image pool, insufficient difficulty, and the lack of human-likeness. In addition, existing related metrics, *e.g.,* FID or SSIM (Wang et al., 2004), are unsuitable since the evaluation of decoding models requires the comparison between two images that could be highly dissimilar. Furthermore, we collected human ratings on the semantic similarities of 1,000 ground-truth (GT) and reconstruction image pairs from 22 evaluators. Using these ratings, we revealed that most existing metrics show a low correlation with human evaluation about the semantic similarity of GT and its brain-decoded reconstruction, with the exception of the EffNet (Tan & Le, 2019) metric. Our finding underscores the urgent need for improved evaluation criteria.

To that end, inspired by the human visual perception process, we propose a new evaluation metric that primarily focuses on the semantic likeness of two images, SEED (**Se**mantic **E**valuation for Visual Brain **D**ecoding). SEED is a combinatorial metric that integrates **two newly proposed** metrics, *Object F1* and *Cap-Sim*, alongside EffNet, a well-established metric, each resembling different stages of the human visual perception pipeline.

More specifically, Object F1 is a metric that aims to identify and capture important elements of the image by automatically detecting and comparing the presence of key objects of the scene using open-vocabulary image grounding models. Cap-Sim is a metric that compares the similarity of the generated captions of two images. This metric captures additional semantic factors that might be overlooked by Object F1, such as *backgrounds, pose*, and *color*, offering a complementary evaluation of the high-level image semantics. EffNet is a widely adopted metric leveraging an ImageNet (Deng et al., 2009) pre-trained EfficientNet (Tan & Le, 2019) model. The metric is known to be particularly well suited to capture the more global and structural aspects of the scene, thus complementing Object F1 and Cap-Sim.

By carefully comparing our proposed and existing metrics with the collected human evaluation results, we show that the two new metrics, Object F1 and Cap-Sim, indeed exhibit strong agreement with human evaluation, and our SEED achieves the highest alignment with human evaluation, compared to all existing metrics. In order to facilitate future research on developing new metrics, we plan to release the human evaluation results.

Furthermore, our evaluation of recent visual brain decoding models with SEED revealed that even the most advanced models frequently fail to accurately reconstruct key objects of interest, often confusing them with similar ones. Even when key objects are correctly identified, the models often struggle to capture semantic details. We believe these findings can provide valuable guidance for advancing research in visual brain decoding.

## 2 BACKGROUND

### 2.1 VISUAL BRAIN DECODING MODELS

Visual brain decoding refers to the task of reconstructing visual stimuli, such as an image, given the brain signals of a human subject that is viewing the said visual stimuli. In the early stages of development of visual decoding models, linear regression-based approaches demonstrated that visual information can be decoded from brain signals (Kamitani & Tong, 2005; Haynes & Rees, 2005). With the development of deep learning techniques, more sophisticated decoding becomes promising, such as GAN (Goodfellow et al., 2020) based visual brain decoding (Seeliger et al., 2018; Ozcelik et al., 2022). Recent decoding models adopt latent diffusion models (Rombach et al., 2022) to produce high-quality decoded images conditioned by brain embeddings or predicted CLIP (Radford et al., 2021) image embeddings from fMRI signals (Scotti et al., 2023; 2024; Wang et al., 2024b;a; Tian et al., 2025; Gong et al., 2025). Instead of freezing the pre-trained diffusion mod-

els, NeuroPictor (Huo et al., 2024) fine-tunes the diffusion model to directly condition the image generation process with brain embeddings.

Beyond the single modality decoding, recent works aim to simultaneously reconstruct the multiple modalities, mainly text and images from a fMRI signals (Mai & Zhang, 2023; Xia et al., 2024b; Shen et al., 2024).

Furthermore, we note that there is a line of work that mainly focuses on the reconstructing textual information from the fMRI signals (Chen et al., 2025a;b), though they are not main focus of our work.

Instead of freezing the pre-trained diffusion models, NeuroPictor (Huo et al., 2024) fine-tunes the diffusion model to directly condition the image generation process with brain embeddings.

## 2.2 CURRENT EVALUATION SCHEMES

Most of the recent decoding literature (Ozcelik & VanRullen, 2023; Scotti et al., 2023; Liu et al., 2025; Scotti et al., 2024; Wang et al., 2024a; Shen et al., 2024; Huo et al., 2024; Wang et al., 2024b; Xia et al., 2024a) mainly focus on the following eight evaluation metrics: PixCorr, SSIM (Wang et al., 2004), AlexNet(2), AlexNet(5) (Krizhevsky et al., 2012), Inception (Szegedy et al., 2015), CLIP (Radford et al., 2021), EffNet (Tan & Le, 2019), and SwAV (Caron et al., 2020).

PixCorr refers to the Pearson correlation between the pixel values of the GT and the reconstruction. SSIM refers to the structural similarity index measure between the GT and the reconstruction.

AlexNet(2), AlexNet(5), Inception, and CLIP refer to the accuracy of two-way identification tasks that use the corresponding feature extractor. Specifically, for every GT embedding, the Pearson correlation with its corresponding reconstruction embedding is compared against its correlation with each other reconstruction embedding in the test set. The percentage of cases in which the GT embedding is closer to its correct reconstruction is reported.

The $n$-way extension of the task utilizing the brain-generated intermediate CLIP embeddings and the GT CLIP image embeddings, known as image/brain retrieval, is also reported in some works (Scotti et al., 2023; 2024; Lin et al., 2022). However, the retrieval tasks are not applicable to models such as NeuroPictor (Huo et al., 2024) as they require the model to generate brain-derived intermediate CLIP image embeddings during the decoding process.

EffNet and SwAV refer to the correlation distance between the GT embedding and the reconstruction embedding, utilizing the corresponding feature extractor.

## 3 ISSUES WITH EXISTING EVALUATION METHODS

### 3.1 EMPLOYMENT OF EXISTING RELATED METRICS

When evaluating visual brain decoding models, it is crucial to measure how closely the reconstruction aligns with the GT, acknowledging potential perceptual and semantic deviations. Unlike typical image generation tasks, which lack a fixed GT, decoding tasks involve a predetermined target. Consequently, standard metrics for image generation, such as FID, are unsuitable, and a measure that directly compares the reconstruction to the known image is required.

In this sense, due to the nature of comparing the similarity of two images, the evaluation of the decoding task more closely resembles traditional image quality assessment, where images are degraded by compression, transmission, or other processes. This is precisely the context for which metrics like SSIM were originally designed, which likely explains why those metrics are widely used for the evaluation of visual brain decoding models.

However, a key distinction lies in the inherent noisiness of decoding, where reconstructions can be perceptually different from the GT while retaining a similar semantic theme. This can result in metrics like SSIM assigning unusually low scores as they are prone to even small distortions, such as translations and rotations (Nilsson & Akenine-Möller, 2020), let alone the larger distortions often found in reconstructions.

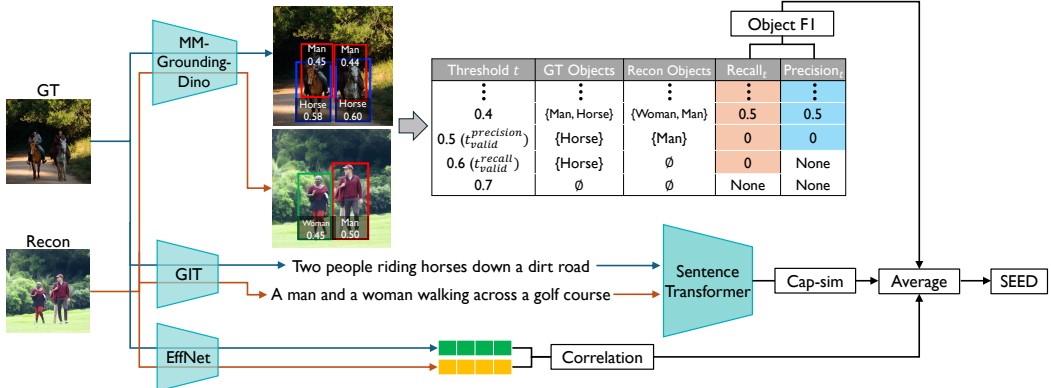

Figure 2: The overall process for calculating SEED.

Consequently, although it might appear that conventional image quality assessment metrics are ideally suited to evaluate decoding models, in practice, they are substantially misaligned from human evaluation, as demonstrated in Sec. 5.1. Therefore, the focus of evaluation should be geared towards assessing the semantic qualities of the reconstructions, due to the noisiness of the decoding process.

## 3.2 TWO-WAY IDENTIFICATION

Two-way identification metrics (AlexNet(2), AlexNet(5), Inception, CLIP) serve a crucial role in the evaluation of decoding models, as they occupy half of the eight-metric evaluation scheme. However, due to their comparative nature, two-way identification metrics contain some inherent flaws. First and foremost, comparing two-way identification scores between models is inappropriate. As each reconstruction is compared against other reconstructions generated by the decoding model, the pool of images each reconstruction is compared against differs for each decoding model. This fact renders the direct comparison of two-way identification scores inappropriate, as each model would be evaluated under different criteria.

Another issue arises from the difficulty, or lack thereof, of the two-way identification task. Since the reconstruction only needs to be closer to the GT than another random example, a reasonable reconstruction easily "wins" the comparison. Due to this, recent decoding models already show near-perfect performance for most two-way identification metrics. This makes it difficult to differentiate the performance between different decoding models and thus calls for a more challenging evaluation task.

## 3.3 LACK OF HUMAN-LIKENESS

Excluding PixCorr and SSIM, all other evaluation metrics rely on abstract features extracted from pre-trained vision models. Consequently, it is difficult to interpret the rationale behind each evaluation from a human perspective, casting doubt on whether they truly align with human perception—especially while under scrutiny. Our human survey findings indeed reveal that most commonly used metrics gauge semantic similarity in ways that deviate notably from human evaluation. Further details are in Sec. 5.1.

## 4 NEW SEMANTIC EVALUATION METHODS

Given the issues outlined in Sec. 3, there is a clear need for evaluation methods that deliver more accurate and generalizable assessments for visual brain decoding. To that end, we borrow inspiration from the human visual attention system to develop new decoding evaluation protocols. Among neuroscientific literature (Jonides, 1983; Treisman, 1998; Zhang, 2019), the common consensus is that visual perception and attention are a two-stage process.

During the first stage, the visual system analyzes basic features of the environment such as color, orientation, and brightness. This process occurs in parallel, simultaneously dividing attention across the entire visual field.

Although the specifics may vary from theory to theory, the second stage of visual attention involves focused attention, which is crucial for binding the separately processed features into coherent, recognizable objects. In this stage, attention is selectively concentrated on specific locations within the visual field. When attention is directed to a particular area, the brain integrates the features present at that location into a unified percept.

We noticed that most existing metrics, especially the ones involving a convolution model, use models that follow a similar process to the first stage, but not the second stage. This observation motivated us to develop two different metrics that each resemble different parts of the second stage, as well as a metric to unify the two stages, namely: Object F1, Cap-Sim, and SEED.

### 4.1 OBJECT F1

We first introduce a metric that focuses on key objects, in order to roughly follow the object-oriented attention mechanism of the second stage of visual attention. Object F1 is a metric that measures the similarity of two images based on object presence; that is, objects present in the GT should also be present in the reconstruction, and objects not present in the GT should also not be present in the reconstruction. Using image grounding models, it is possible to automatically detect the objects present in both images and quantify the aforementioned criterion into two proposed metrics: Object Recall and Object Precision.

We first run all GT and reconstructed images through an image grounding model and obtain the detection results. The results should contain the list of detected objects with information such as the category and the confidence value for each object. Given a confidence threshold $t$, which is the threshold used to determine whether an object is "detected," we define two preliminary metrics for each image: Object Recall$_t$ and Object Precision$_t$.

Object Recall$_t$ measures the proportion of the object categories from the GT that are also present in the reconstruction. This measures the proportion of objects that are successfully "recalled" in the reconstruction, formulated as:

$$\text{Object Recall}_t := \frac{\text{\# of categories in both GT and recon}}{\text{\# of categories in GT}} \tag{1}$$

Similarly, Object Precision$_t$ measures the proportion of the object categories from the reconstruction that are also present in the GT. This essentially measures the "precision" of the objects in the reconstruction, formulated as:

$$\text{Object Precision}_t := \frac{\text{\# of categories in both GT and recon}}{\text{\# of categories in recon}} \tag{2}$$

During the process, we apply the same threshold value to the GT and reconstruction to ensure the ideal reconstruction (i.e., reconstruction identical to the GT) obtains the best possible score. For simplicity, if multiple objects of the same category are present in an image, we only consider the object with the highest score, as we only check for the existence of each object category.

To remove the reliance on a threshold hyperparameter, we calculate Object Recall$_t$ and Object Precision$_t$ while moving the threshold, $t$, between 0 and 1 and obtain the averaged values:

$$\text{Object Recall} := \frac{1}{t_{\text{valid}}^{\text{recall}}} \int_0^{t_{\text{valid}}^{\text{recall}}} \text{Object Recall}_t \, dt$$

$$\text{Object Precision} := \frac{1}{t_{\text{valid}}^{\text{precision}}} \int_0^{t_{\text{valid}}^{\text{precision}}} \text{Object Precision}_t \, dt \tag{3}$$

where $t_{\text{valid}}^{\text{recall}}, t_{\text{valid}}^{\text{precision}}$ are cutoff thresholds, corresponding to the highest confidence value present in the GT and reconstruction, respectively. The threshold is cut off in such a way since there would be no detected objects for higher threshold values.

The final evaluation metric, Object F1, is the harmonic mean of the averaged Object Recall and Object Precision:

$$\text{Object F1} := \frac{2}{\text{Object Recall}^{-1} + \text{Object Precision}^{-1}} \tag{4}$$

The threshold-averaging scheme has the added benefit of penalizing reconstructions with objects far apart from the GT in terms of confidence, as those objects would be marked as incorrect during the intermediate threshold values. This trait is beneficial for evaluating decoding models, as they often generate distorted objects (Scotti et al., 2024) that tend to show lower confidence values than their GT counterparts.

We note that the proposed Object F1 fundamentally differs from the Average Precision (AP) in object detection. AP evaluates *detection models* by comparing bounding boxes based on IoU for a *single image*, whereas Object F1 measures similarity of *two images* based on object existence, *independent from IoU*.

To calculate Object F1, we employ MM-Grounding-DINO (Zhao et al., 2024) to detect 82 object categories; the full list of categories is available in Sec. B.1. For Object Recall and Object Precision, to approximate Eq. 3, we move the threshold $t$ from 0 by increments of 0.01, up to the cutoff thresholds, and average the values.

## 4.2 CAP-SIM

Similar to how Object F1 emulates the object-oriented attention mechanism of the second stage of visual attention, we introduce a metric inspired by the subsequent process within the same stage that identifies and binds relevant features. Cap-Sim is a metric that measures the similarity between captions generated by image captioning models for each GT and reconstruction pair. Instead of relying on abstract features generated by vision models, this approach emphasizes semantic qualities expressible by natural language since the images are essentially "compressed" into text before being compared. This method allows us to evaluate semantic factors that are hard to identify through the existence of objects, such as the background information or attributes of the detected object (pose, color, etc.). Furthermore, caption-based evaluation provides an interpretable assessment, as captions are human-readable and closely align with how people describe visual content (He et al., 2019).

Formally, Cap-Sim is formulated as:

$$\text{Cap-Sim} := \cos(e_{\text{text}}(c(I_{GT})), e_{\text{text}}(c(I_{recon}))) \tag{5}$$

where $I_{GT}$ and $I_{recon}$ are GT and reconstructions, respectively. The functions $e_{\text{text}}(\cdot)$ and $c(\cdot)$ denote text encoder and caption generator, respectively, for which we use Sentence Transformer (Reimers & Gurevych, 2019) and GIT (Wang et al., 2022). To the best of our knowledge, we note that caption-based evaluation of image similarity has not been previously proposed, despite its simplicity.

## 4.3 SEED

Building on these metrics, we aim to construct a unified evaluation framework that captures the complementary aspects of human visual attention, each modeled by the individual metrics, and serves as a reliable standard for assessing decoding models. To this end, we introduce **Se**mantic **E**valuation for Visual Brain **D**ecoding (SEED), a composite metric that integrates Object F1, Cap-Sim, and $\overline{\text{EffNet}}$.

Note that $\overline{\text{EffNet}}$ is a slightly modified metric by calculating **correlation**, not **correlation distance**, converting it into a higher-is-better metric like the other two;

$$\overline{\text{EffNet}} := corr(e_{\text{img}}(I_{GT}), e_{\text{img}}(I_{recon})) \tag{6}$$

where the function $e_{\text{img}}(\cdot)$ is the image encoder, EffNet.

The overall procedure to compute SEED and its components for a given image pair is depicted in Fig. 2. We simply take the average of the three metrics to calculate SEED:

$$\text{SEED} := (\text{Object F1} + \text{Cap-Sim} + \overline{\text{EffNet}}) \,/\, 3 \tag{7}$$

## 4.4 HUMAN EVALUATION OF IMAGE SIMILARITY

We collected 5-point Likert scale ratings from 22 human evaluators to assess the alignment of current evaluation metrics with human evaluation. They assessed both the semantic and perceptual similarity between GT and their reconstructions for 1,000 test set images in Natural Scenes Dataset (NSD) (Allen et al., 2022) used by Scotti et al. (2024), where the reconstructions were generated by the MindEye2 model released by the original author, with 250 reconstructions sequentially sampled from each of the four subjects (subject 1, 2, 5, and 7), following the order: the first 250 from subject 1, the next 250 from subject 2, and so on. The intraclass correlation (ICC(2, n)) (Koch, 2006) between the human evaluation results is 0.84 ($p = 0$), indicating a sufficiently high inter-rater agreement. Further detailed information on the collection of human ratings is provided in Sec. A, and we will release the survey results to facilitate future research on similar topics.

# 5 EXPERIMENTAL RESULTS

## 5.1 ALIGNMENT WITH HUMAN EVALUATION

Table 1: The meta-evaluation results on NSD with MindEye2. The best results are **bolded**. $\overline{\text{SwAV}}$ was calculated similarly to Eq. 6.

| Metric | Pairwise Acc. | Kendall | Pearson |
|--------|---------------|---------|---------|
| PixCorr | 53.8% | .075 | .117 |
| SSIM | 54.5% | .090 | .112 |
| AlexNet(2) | 55.0% | .185 | .187 |
| AlexNet(5) | 49.5% | .236 | .258 |
| Inception | 63.8% | .330 | .475 |
| CLIP | 66.4% | .368 | .436 |
| $\overline{\text{EffNet}}$ | 78.0% | .559 | .748 |
| $\overline{\text{SwAV}}$ | 69.7% | .394 | .576 |
| Object F1 | 75.8% | .516 | .708 |
| Cap-Sim | 73.8% | .477 | .666 |
| SEED | **81.0%** | **.621** | **.813** |

Table 2: The meta-evaluation results of reconstructions of the GOD dataset with Mind-Vis. The best results are **bolded**.

| Metric | Pairwise Acc. | Kendall | Pearson |
|--------|---------------|---------|---------|
| PixCorr | 51.3% | .029 | .078 |
| SSIM | 49.2% | -.013 | -.103 |
| AlexNet(2) | 66.0% | .377 | .492 |
| AlexNet(5) | 65.8% | .423 | .445 |
| Inception | 62.6% | .324 | .356 |
| CLIP | 63.2% | .338 | .309 |
| $\overline{\text{EffNet}}$ | 72.5% | .453 | .661 |
| $\overline{\text{SwAV}}$ | 68.6% | .376 | .498 |
| Object F1 | 66.0% | .322 | .431 |
| Cap-Sim | 68.7% | .376 | .577 |
| SEED | **73.7%** | **.477** | **.706** |

Following Lin et al. (2024), we adopt pairwise accuracy (Deutsch et al., 2023), Kendall's Tau-b, and Pearson correlation to meta-evaluate each metric based on the human ratings of the semantic similarity between images. We meta-evaluated eight metrics widely used in prior works (Scotti et al., 2023; 2024; Wang et al., 2024a;b; Tian et al., 2025). Additionally, we explored alternative approaches for measuring the semantic similarity between images based on visual question answering models, detailed in Sec. C.2.

The meta-evaluation results, presented in Tab. 1, indicate that most existing metrics exhibit low correlation with human evaluation, except for $\overline{\text{EffNet}}$. Furthermore, the alternative approaches do not perform as effectively as Object F1 or Cap-Sim. Notably, SEED achieves the highest agreement with human evaluation with statistical significance. To assess the statistical significance of the improvement of SEED over $\overline{\text{EffNet}}$, which shows strong alignment among existing metrics, We performed bootstrapping along the evaluator axis (sample size = 22) for 1,000 iterations and computed the confidence intervals of the differences in each meta-evaluation metric between SEED and $\overline{\text{EffNet}}$. The 95% confidence intervals for pairwise accuracy, Kendall's Tau-b, and Pearson correlation were $[0.03, 0.07]$, $[0.02, 0.04]$, and $[0.04, 0.08]$, respectively, all of which do not include zero. These results indicate that the performance improvement of SEED over $\overline{\text{EffNet}}$ is statistically significant.

We note that the combination of the three metrics is essential to achieve the highest alignment with human evaluations. A detailed analysis is provided in Sec. C.3.

## 5.2 ROBUSTNESS OF SEED

Since several factors in SEED may influence the evaluation process, we conduct experiments to examine its robustness under different scenarios.

**Robustness to dataset and decoding model.** One major factor affecting meta-evaluation would be the choice of dataset and decoding model that serves as the evaluation target. To perform meta-evaluation on a different setting, we collected human evaluations from 10 student volunteers for 50 reconstructions generated by Mind-Vis (Chen et al., 2023) on the General Object Decoding (GOD) dataset (Horikawa & Kamitani, 2017). The ICC values for semantic similarity was 0.93 ($p = 0$), indicating high agreement among raters. We used the full list of 50 test set class names to compute Object F1. As shown in Tab. 2, SEED again achieved the highest alignment with human evaluation, demonstrating that it generalizes well across datasets and decoding models.

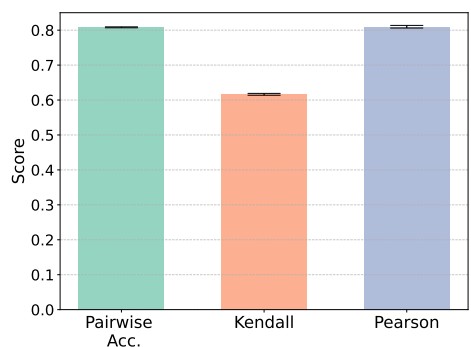

**Robustness to the choice of off-the-shelf models.** We next evaluated whether SEED's performance depends on the specific choice of image grounding model, caption generator $c(\cdot)$, or text encoder $e_{\text{text}}(\cdot)$. We substituted the original components with Yolo-World (Cheng et al., 2024) for image grounding, BLIP-2 (Li et al., 2023) for caption generation, and Qwen3-Embedding-0.6B (Zhang et al., 2025) for text encoding. Meta-evaluation results across all eight model combinations are summarized in Fig. 3. The barplots indicate that performance differences across all choices are minimal, confirming that SEED is robust to the selection of these off-the-shelf models.

Figure 3: Meta-evaluation results with different choices of off-the-shelf models.

## 5.3 ANALYSIS OF WORST-CASE JUDGMENTS

To understand why SEED improves upon its components, we present case studies of the "worst-case judgments" for each component of SEED, despite their high agreement with human evaluation. In this context, "worst-case judgments" refer to images whose metric-based ranking differs significantly from the human evaluation ranking. Rankings were computed from each metric's numeric scores and from human ratings, where human ratings were normalized per evaluator and then averaged per image. The examples shown in Fig. 4 are chosen among the worst-case judgments for each metric, where the other two metrics made a human-aligned decision, which somewhat mitigates the discrepancy. Additional examples are available in Sec. D.3.

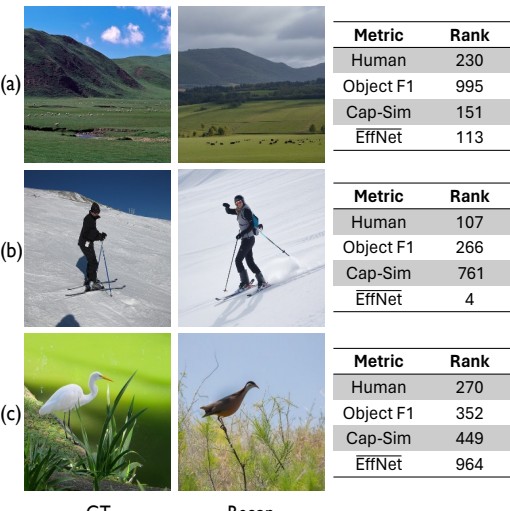

Figure 4: Visualizations (out of 1000 pairs) of worst-case judgments for (a) Object F1, (b) Cap-Sim, and (c) EffNet.

Fig. 4 (a) shows a case where Object F1 significantly deviates from human evaluation and other metrics by assigning a score of 0. This disparity arises because Object F1 fails to capture global scene information, relying solely on detected animals (*sheep* in the GT and *cow* in the reconstruction).

Fig. 4 (b) shows a case where Cap-Sim significantly deviates from the others, where the caption generated by GIT is [*A man on skis standing on a snowy hill.*] and [*A woman on skis is waving while skiing.*] for the GT and the reconstruction, respectively. The low similarity likely results from the change of gender or the described action, despite other metrics as well as humans assigning a high similarity.

Fig. 4 (c) shows a case where $\overline{\text{EffNet}}$ significantly deviates from the others. Although it is difficult to pin down the exact reason, one possible explanation is the fact that the two images have different

ImageNet Top-1 predictions from the EffNet model: *American egret* for the GT and *Coucal* for the reconstruction. We hypothesize that the EffNet tends to over/underestimate the correlation between two images with the same/different class predictions.

To validate this suspicion, we compared the average z-normalized EffNet and the human semantic evaluation scores of the image pairs with the same/different EffNet ImageNet Top-1 predictions. For images from the same class, EffNet yields an average score of 0.755, whereas human evaluators score 0.313 on average. For images of different classes, the average scores are -0.333 for EffNet and -0.138 for humans. This indicates that EffNet produces overestimated assessments, depending on the ImageNet classes, and we believe this explains EffNet's low correlation for cases like Fig. 4 (c).

## 5.4 FAILURE MODE DISCOVERY

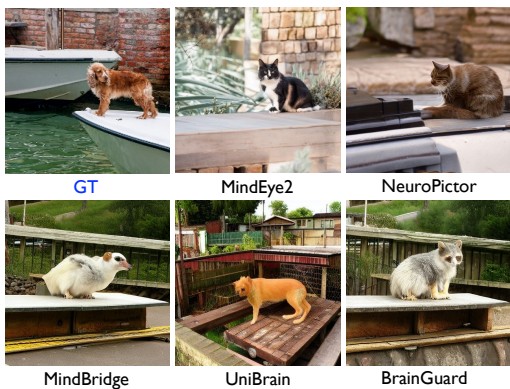

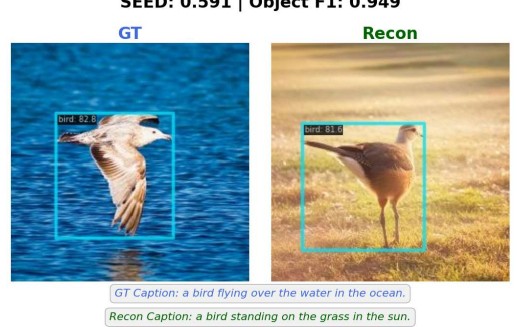

Figure 5: Examples of the semantic near-miss phenomenon.

Figure 6: An example of reconstruction which captures objects correctly but misses semantic details.

**Semantic near-miss phenomenon.** One common failure mode of current decoding models is the *semantic near-miss* phenomenon, in which the reconstruction misrepresents the specific object category from the GT, yet still captures the broader supercategory. For example, if the GT contains a *dog*, the reconstruction might include a *cat* or other animals (See Fig. 5.). While this *cat* is in the wrong category, it remains within the correct supercategory, *animal*.

We quantify this by re-using the object detection pipeline used in Object F1. We calculate the Object Recall (Eq. 1) and the *Relaxed* Object Recall, which measures the proportion of the object categories from the GT where its supercategory (instead of the specific category) is present in the reconstruction. The gap between those two represents the rate of the semantic near-miss phenomenon.

We computed the semantic near-miss rate of salient object categories (Xia et al., 2024b) at a confidence threshold of 0.3 for five existing decoding models in Sec. E, and observed rates ranging from 17.5% to 20.6%. Such a high incidence indicates that current decoding models often struggle with fine-grained object differentiation, capturing only coarse semantic details.

**Captured objects while missing semantic details.** We identify another failure mode in which the model reconstructs the main objects but overlooks crucial semantic details. To analyze this, we focus on reconstructions with high Object F1 but low overall SEED, specifically those satisfying Object F1 $> 0.7$ and Object F1 $-$ SEED $> 0.2$. While the exact thresholds are somewhat arbitrary and can be varied, our goal here is *not to fixate on specific cutoff values* but *to demonstrate how such criteria enable systematic identification of failure modes*. This criterion isolates cases where low Cap-Sim and EffNet scores reduce the SEED average. Such cases indicate that while the model successfully reconstructs objects, it often fails to capture other details such as backgrounds, pose, or color. Fig. 6 illustrates one such example, where the reconstruction correctly captures a *bird* but fails to reconstruct the background as well as its pose.

Using this criterion, we measured the proportion of reconstructions. The ratio ranges from 8.3% to 10.7% across the five decoding models evaluated in Sec. E, suggesting that a sizable fraction

of reconstructions, while correctly identifying the main objects, still fail to recover fine-grained semantic details.

**Potential remedies.** While we do not propose solutions for these failure modes, we believe that our findings suggest several promising research directions. First, more systematic error analysis with SEED could provide actionable guidance for data collection. For example, if a model reliably reconstructs objects but frequently mismatches backgrounds, this would suggest collecting images with greater background diversity. Similarly, to address the semantic near miss phenomenon, one could gather datasets containing images with subtle differences between them. Second, training strategies could aim to disentangle object reconstruction from semantic detail reconstruction. Most current decoding models use CLIP image embeddings as regression targets, which may conflate these two aspects and contribute to the failures. Future methods may therefore benefit from decoupling object-level supervision from supervision for other details.

## 6 CONCLUSION & LIMITATIONS

In this work, we introduce **SEED**, a novel framework designed to assess the semantic decoding performance of decoding models. Through comprehensive experiments, we show that existing evaluation metrics often diverge from human judgments, whereas our proposed metric exhibits stronger alignment and improved reliability.

Our results reveal a growing mismatch between the goals of modern visual brain decoding and the metrics currently used to evaluate it. Although recent diffusion-based models can achieve near-perfect scores on traditional identification metrics and display high similarity scores, our human-aligned analyses show that these models often overlook substantial semantic errors, including missing objects, incorrect categories, and failures to capture contextual details, which are overlooked by traditional metrics. This indicates that the field may be overestimating progress due to evaluation tools that no longer reflect the true complexity of the task.

SEED addresses this gap by providing a more human-consistent measure of semantic fidelity, integrating object-level, caption-level, and other fine-grained semantic cues. Beyond offering a more reliable evaluation metric, SEED reveals distinct failure modes, such as semantic near-misses and losses of fine detail, thereby enabling more targeted model development.

More broadly, our findings highlight that as decoding models mature, so too must our evaluation practices. We hope that SEED encourages the community to adopt richer, human-aligned evaluation frameworks and to develop models that capture objects, attributes, and other semantic details in a more faithful and robust manner.

**Limitations and future work.** Nonetheless, our approach has its limitations. As SEED depends on the off-the-shelf models, SEED may inherit systematic errors from the existing models. One such example is provided in Sec. D.2, where all metrics of SEED fail to make a human-aligned judgment when an unusual or malformed image is given as the reconstruction, which in turn leads to the failure of SEED. Training evaluation models or devising metrics that are more robust to these scenarios could be a promising future direction.

In addition, because SEED was designed with a stronger emphasis on evaluating image semantics, it may become less effective once precise assessment of perceptual details is required as brain decoding technology matures. While we currently regard accurate semantic decoding as the higher priority, we expect that, as models improve and reliably capture high-level semantics, the focus will naturally shift toward perceptual fidelity. At that stage, an evaluation method better suited to detecting fine-grained perceptual aspects should be introduced.

## REPRODUCIBILITY STATEMENT

For the reproducibility of our study, we detailed the model used for computation of SEED in Sec. 4 and how to compute SEED. In addition, our code and the human evaluation results are available at `https://github.com/Concarne2/SEED`.

ACKNOWLEDGMENT

This work was supported in part by National Research Foundation of Korea (NRF) grant [No. 2021R1A2C2007884, No. RS-2025-02263628, No. RS-202300265406], the Institute of Information & communications Technology Planning & Evaluation (IITP) grants [RS-2021-II212068, RS-2022-II220113, RS-2022-II220959, RS-2021-II211343], the BK21 FOUR Education and Research Program for Future ICT Pioneers (Seoul National University), funded by the Korean government (MSIT), the Ministry of Education [RS-2024-00435727], the National Supercomputing Center [KSC-2025-CRE-0340], the U.S. Department of Energy's ASCR Leadership Computing Challenge [m4750-2024], and Hyundai Motor Chung Mong-Koo Foundation.

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

# SEED: Towards More Accurate Semantic Evaluation for Visual Brain Decoding

## Appendix

### THE USE OF LARGE LANGUAGE MODELS (LLMS)

We utilized LLMs for the purpose of polishing our manuscript only.

## A   COLLECTION OF HUMAN EVALUATIONS

We used the Amazon Mechanical Turk (MTurk) platform as well as additional student evaluators to collect human ratings on the semantic and perceptual similarity between GT and its reconstruction. A screenshot of the survey window is shown in Fig. 7.

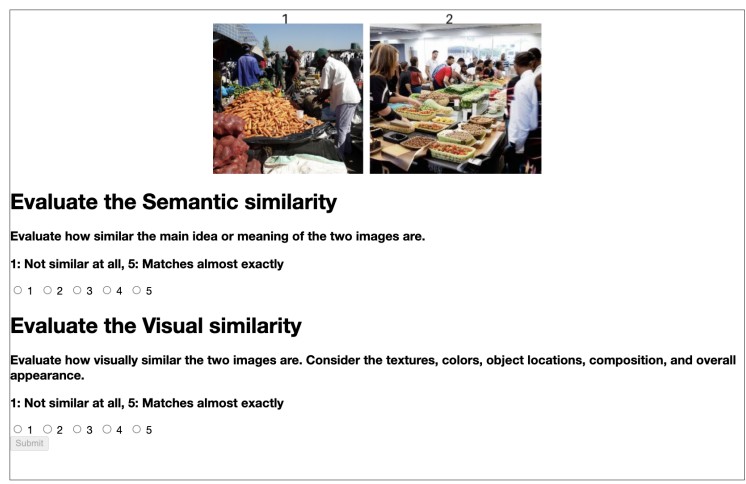

Figure 7: A screenshot of our Amazon MTurk survey window.

Referring to Otani et al. (2023), we applied the following filter for worker requirements when creating the MTurk project: 1) Master: Good-performing and granted AMT Masters. Each annotator was paid $0.03 for evaluating the semantic and perceptual similarity of a single pair of GT and its reconstruction image. We gathered a total of 22 ratings for each of the 1,000 pairs.

The intraclass correlation (ICC(2, n)) (Koch, 2006) for the perceptual similarity evaluation results was 0.79 with $p = 0$, which indicates high inter-rater agreement.

## B   CHOOSING CANDIDATE OBJECT CATEGORIES FOR OBJECT DETECTION

### B.1   FULL LIST OF OBJECT CATEGORIES

The list of object categories, which was used for object detection, is composed of 80 COCO categories plus 2 additional human categories (*man* and *woman*). The resulting 82 categories can be further classified into 30 "Salient" and 52 "Inconspicuous" objects as per Xia et al. (2024b).

The 30 salient objects are: [person, man, woman, bird, cat, dog, horse, sheep, cow, elephant, bear, zebra, giraffe, bicycle, car, motorcycle, airplane, bus, train, truck, boat, bench, chair, couch, bed, dining table, toilet, sink, refrigerator, clock]

The 52 inconspicuous objects are: [traffic light, fire hydrant, stop sign, parking meter, backpack, umbrella, handbag, tie, suitcase, frisbee, skis, snowboard, sports ball, kite, baseball bat, baseball glove, skateboard, surfboard, tennis racket, bottle, wine glass, cup, fork, knife, spoon, bowl, banana,

apple, sandwich, orange, broccoli, carrot, hot dog, pizza, donut, cake, potted plant, tv, laptop, mouse, remote, keyboard, cell phone, microwave, oven, toaster, book, vase, scissors, teddy bear, hair drier, toothbrush].

### B.2 CHOOSING CATEGORIES WITH VLM

The rapid development of vision-language models (VLM) made us wonder if the process of choosing object categories could be delegated to VLMs instead of using a fixed set of objects. To answer this question, we use an open-sourced Qwen2.5-VL-7B-Instruct (Bai et al., 2025) model to extract the object categories instead of using the aforementioned 82. We gave the model each GT and reconstruction image separately; we experimented with different text prompts, but the following was the most effective: *Generate a list of objects and background features that are present in the image. Only answer in a comma-separated list of objects. Do not include any other text or explanation.* With the extracted object categories, we calculated the Object Recall with the categories of the GT and the Object Precision with the categories of the reconstruction image, separately for each image pair. Compared to the fixed list of 82 categories, which is the one used in the manuscript, this strategy performed slightly worse, although still significantly outperformed existing metrics.

Table 3: The meta-evaluation results while using a fixed set of 82 categories versus VLM-generated object categories.

| Metric | Pairwise Acc. | Kendall | Pearson |
|---|---|---|---|
| Object F1 | **75.8**% | **.516** | **.708** |
| Object F1 (VLM) | 73.7% | .473 | .658 |
| SEED | **81.0**% | **.621** | **.813** |
| SEED (VLM) | 80.4% | .607 | .800 |

## C ADDITIONAL ANALYSES

### C.1 INCORPORATION OF LOCATION, SIZE, AND NUMBER INFORMATION

Table 4: The meta-evaluation results of Object F1 with incorporation of additional information.

| Options | | | | Pairwise Acc. | Kendall | Pearson |
| Existence | Size | Location | Number | | | |
|---|---|---|---|---|---|---|
| ✓ | | | | 75.8% | .516 | .708 |
| ✓ | ✓ | | | 75.8% | .517 | .709 |
| ✓ | | ✓ | | **75.9**% | **.517** | **.710** |
| ✓ | | | ✓ | 74.7% | .493 | .648 |

We incorporate location, size, and number information into Object F1 to determine whether each factor contributes to the improvement of alignment with human evaluations, as outlined below:

**Size weighting** We weight object categories based on their bounding box size, with larger sizes receiving higher weights. An object that fills the entire image would be weighted twice as much as an object with zero area, with scaling linearly.

**Location weighting** We weight object categories based on their proximity to the center of the image, with objects closer to the center receiving higher weights. An object at the center would be weighted twice as much as an object at the edge of the image, with scaling linearly.

**Number count** During recall and precision calculation, each object category receives partial credit if the number of detected object categories is either underestimated or overestimated, depending on the error.

The results are summarized in Tab. 4. Since none of these weighting schemes seemed to improve the metric, they were not included in the final version in order to avoid needlessly complicating the metric.

## C.2 ADDITIONAL RESULTS OF SEC. 5.1

Table 5: The meta-evaluation results of each metric. The best results are **bolded**.

| Metric | Pairwise Acc. | Kendall | Pearson |
|---|---|---|---|
| Object F1 | 75.8% | .516 | .708 |
| Cap-Sim | 73.8% | .477 | .683 |
| $\overline{\text{EffNet}}$ | 78.0% | .559 | .748 |
| Object F1+ Cap-Sim | 78.3% | .566 | .768 |
| Object F1+ $\overline{\text{EffNet}}$ | 80.1% | .602 | .794 |
| Cap-Sim+ $\overline{\text{EffNet}}$ | 79.2% | .583 | .787 |
| BLIP VQAScore | 71.3% | .427 | .566 |
| GIT VQAScore | 71.7% | .434 | .574 |
| SEED | **81.0%** | **.621** | **.813** |

We present additional meta-evaluation results for all possible combinations of components of SEED in Tab. 5. In addition, we explored alternative options for measuring the semantic similarity: CLIP-FlanT5 VQA scores (Lin et al., 2024) with BLIP/GIT generated captions for GT images. Indeed, it can be observed that SEED demonstrates the best agreement with human evaluations.

## C.3 COMBINATION OF EVALUATION METRICS

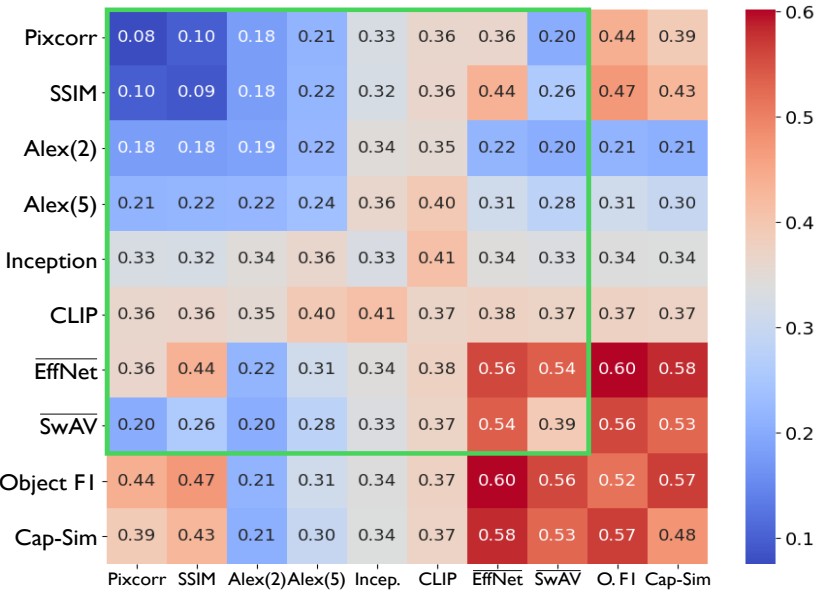

Figure 8: The heatmap of correlations between metric combinations and human evaluation, measured by Kendall's Tau-b. The green outline indicates combinations within current metrics.

To investigate possible candidate metrics that could be included in SEED, we computed the correlation with human evaluations for each possible metric combination, as shown in Fig. 8. The combination is calculated by simply averaging the two metrics. The highest-performing metrics come from the combination of Object F1, Cap-Sim, and $\overline{\text{EffNet}}$, with each combination outperforming the individual components. This result naturally prompts the combination of those three to obtain SEED.

One interesting observation is that it is impossible to create a superior evaluation metric by combining existing metrics; all possible combinations within existing metrics are not better than standalone $\overline{\text{EffNet}}$. A better metric emerges *only when combined with Object F1 or Cap-Sim*. We believe that

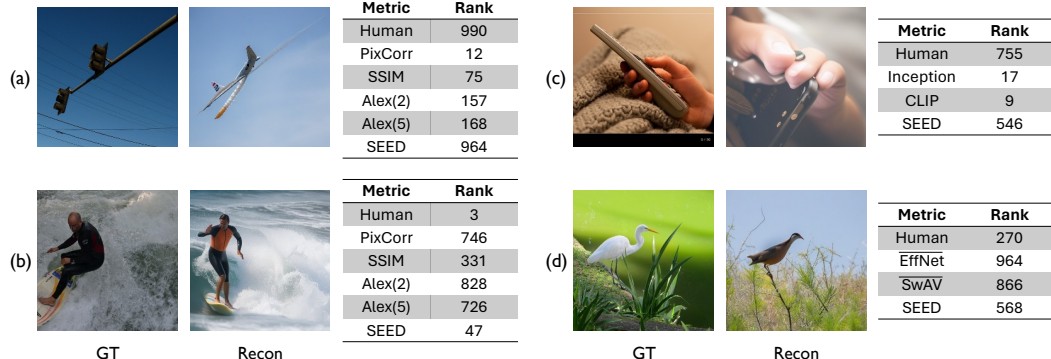

Figure 9: Examples of worst-case judgments for other metrics

this is one indirect evidence that our proposed metrics evaluate the reconstructions from a different angle from EffNet, making it possible for them to work as a complementary metric for each other.

# D    ADDITIONAL EXAMPLES AND ANALYSIS OF WORST-CASE JUDGMENTS

## D.1    WORST-CASE JUDGMENTS FOR OTHER METRICS

Discussions of worst-case judgments in Sec. 5.3 were focused on individual metrics of SEED in order to provide insight as to why SEED performed better than its components. In Fig. 9, we provide some worst-case judgments for the existing metrics (PixCorr, SSIM, AlexNet, Inception, and CLIP) to analyze cases where those metrics make mistakes and how SEED might improve upon them.

Fig. 9 (a) and (b) represent cases where the four low-level metrics, PixCorr, SSIM, Alex(2), and Alex(5), either overestimates or underestimates the similarity of the two images. It is fairly straight-forward to see why those misjudgments came to be for these low-level metrics: for (a), we can see the reconstruction put a malformed airplane in place of the traffic light while the general shape and the background matches the GT. This semantic mismatch made humans as well as SEED to rank this pair very low, while the metrics ranked this pair relatively high since the general shape and color of these match pretty well. For (b), we can see both pictures depict a surfing man, while the specific shape of the waves and the general color tone of the two quite differ. This probably led to humans and SEED to highly rank this pair while the low-level metrics to generally rank this pair low.

For the high-level metrics, it was more difficult to pinpoint the causes for any mistakes or find a reliable pattern between the mistakes, compared to the low-level metrics, due to their abstract nature. Nevertheless, in Fig. 9 (c) and (d), we show the worst-case judgments for the four high-level metrics, further grouped based on their evaluation method. (c) shows a worst-case judgment for the 2-way identification methods, Inception and CLIP. We can see that the reconstruction depicts a slightly disfigured hand, while the object held by the hand was changed from a remote control to a smartphone. This difference likely led to humans and SEED to not favor the reconstruction, while Inception and CLIP might have overvalued the reconstruction since it still features a hand. (d) shows a worst-case judgment for the two correlation distance metrics, $\overline{\text{EffNet}}$ and $\overline{\text{SwAV}}$, which is an example brought from Fig. 4 (c). We can see that $\overline{\text{SwAV}}$ made a misjudgment similar to $\overline{\text{EffNet}}$. We suspect the cause for this mistake is similar, since SwAV was also trained using ImageNet.

## D.2    WORST-CASE JUDGMENTS FOR SEED

Of course, SEED is not a flawless evaluation metric. SEED has the potential to make a misjudgment when its three elements all make a misjudgment for one reason or another, which is displayed in Fig. 10. Here we can see the GT is an image with a person holding a red umbrella, while the recon-sturction is a slightly ambiguous image with a yellow/blue umbrella-like object on top of a wooden object, with a lake on the background. Humans slightly favored this reconstruction since the general pose of the image is similar and the umbrella was somewhat reconstructed. However, all elements

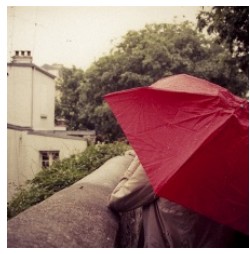 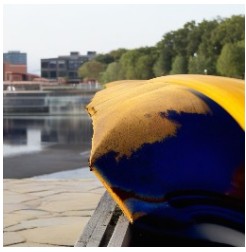

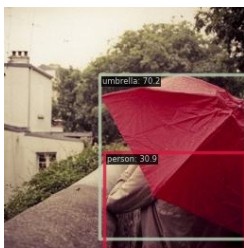 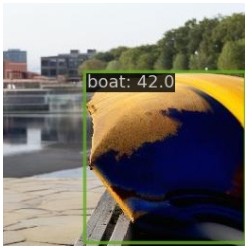

| Metric | Rank |
|--------|------|
| Human | 314 |
| Object F1 | 793 |
| Cap-Sim | 847 |
| EffNet | 773 |
| SEED | 849 |

a person holding a red
umbrella over a wall.

a yellow and blue
towel sitting on top
of a wooden bench.

GT                    Recon

Figure 10: Example of worst-case judgment for SEED

of SEED undervalued this reconstruction, which consequently led to SEED to also undervalue the reconstruction. If we look into the reason, Object F1 gave a poor score since the person from the GT is missing while the yellow/blue umbrella was detected as a boat instead, probably due to the wooden protrusion and the watery background. Cap-Sim gave a poor score for a similar reason; the person was missing from the reconstruction caption, the yellow/blue umbrella was identified as a towel, and the wooden bench was added to the caption. While it is difficult to know the rationale, EffNet gave a poor score, presumably due to the background and the color of the umbrella of the reconstruction being different.

As illustrated by this example, SEED has a chance to fail when the reconstruction is distorted or has some unusual features. This essentially puts the models in an out-of-distribution setting, and they may make a decision that is not aligned with a typical human judgment. Improving the object grounding model or the image captioning model of SEED to better generalize to these distorted images, or advancing the brain decoding models to not produce distorted images in the first place would help in these scenarios.

### D.3 ADDITIONAL WORST-CASE JUDGMENTS FOR SEED ELEMENTS

Here, we present additional examples of the worst-case judgments discussed in Sec. 5.3.

Table 6: Evaluation results with pre-trained models provided by authors. SNM represents the proportion of "semantic near-miss." SDM quantifies the proportion of "semantic detail misses", defined as the fraction of cases with Object F1 $> 0.7$ and Object F1 $-$ SEED $> 0.2$. *MindEye2 was evaluated with 18 additional images, following the original work.

| Method | High-Level | | | | Object F1 ↑ | Cap-Sim ↑ | SEED ↑ | SNM | SDM |
|--------|-----------|------|---------|-------|-------------|-----------|--------|-----|-----|
| | Incep ↑ | CLIP ↑ | EffNet ↓ | SwAV ↓ | | | | | |
| MindEye2* (Scotti et al., 2024) | 95.1% | 93.2% | .617 | .340 | .517 | .542 | .481 | .175 | .107 |
| NeuroPictor (Huo et al., 2024) | 94.6% | 93.5% | .637 | .350 | .481 | .512 | .452 | .191 | .097 |
| MindBridge (Wang et al., 2024a) | 92.6% | 94.7% | .702 | .411 | .440 | .470 | .403 | .203 | .083 |
| UniBrain (Wang et al., 2024b) | 92.3% | 93.7% | .695 | .406 | .453 | .488 | .415 | .206 | .093 |
| BrainGuard (Tian et al., 2025) | 94.8% | 94.8% | .645 | .374 | .489 | .525 | .456 | .192 | .092 |

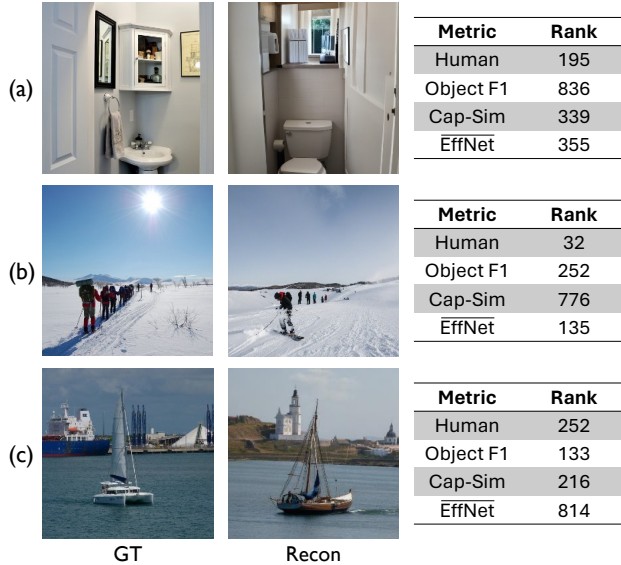

| Metric | Rank |
|--------|------|
| Human | 195 |
| Object F1 | 836 |
| Cap-Sim | 339 |
| EffNet | 355 |

| Metric | Rank |
|--------|------|
| Human | 32 |
| Object F1 | 252 |
| Cap-Sim | 776 |
| EffNet | 135 |

| Metric | Rank |
|--------|------|
| Human | 252 |
| Object F1 | 133 |
| Cap-Sim | 216 |
| EffNet | 814 |

GT          Recon

Figure 11: Additional examples of worst-case judgments

Fig. 11 (a) illustrates a case where Object F1 significantly deviates from human evaluation, assigning a score of 0. This discrepancy arises because the detected category from the GT is *Sink*, while the detected category from the reconstruction is *Toilet*. Since Object F1 evaluates similarity based solely on the presence of the detected category, it assigns a zero score, despite the reconstruction successfully generating an image that represents the concept of a restroom.

Fig. 11 (b) illustrates a case where Cap-Sim assigns a low similarity score between two images. The captions generated by GIT for the GT and the reconstruction are [*A group of people walking across a snow covered field.*] and [*A person riding skis on a snowy surface.*], respectively. This low similarity is likely due to the different actions that people in the image are taking, despite human and other evaluation metrics considering them similar.

Fig. 11 (c) presents a case where the EffNet metric produces an extremely low correlation between two images. The ImageNet Top-1 predictions for the GT and the reconstruction are *Container ship* and *Traffic light*, respectively. This example highlights how EffNet can yield an *incorrect* evaluation due to *misclassification*.

Although the main objects in both images resemble a yacht-like boat, EffNet assigns them to different classes. We believe this occurs because the class *yacht* is not included in the 1,000 ImageNet categories. Consequently, EffNet predicts the GT as a *Container ship*, likely focusing on the ship behind the yacht, while misclassifying the reconstruction as *Traffic light*, a completely irrelevant class.

## E    RE-EVALUATION OF EXISTING DECODING MODELS

We report the performance of existing visual decoding models evaluated with SEED in Tab. 6. We report the evaluation results of five recent decoding models: MindEye2, NeuroPictor, MindBridge, UniBrain, and BrainGuard. We directly evaluated the pre-trained models provided by the authors of each work. The evaluation metrics consist of four existing evaluation metrics alongside our proposed Object F1, Cap-Sim, SEED, and the semantic near-miss rate. Note that MindEye2 was evaluated with 18 additional test image pairs as per the original work due to the sequential disclosure of the NSD dataset.

