# OpenReview forum: "SEED: Towards More Accurate Semantic Evaluation for Visual Brain Decoding"
_ICLR.cc/2026/Conference — ICLR 2026 Poster_

### Official Review · Reviewer_cg7v · 2025-10-14

**Soundness:** 4
**Presentation:** 3
**Contribution:** 3
**Rating:** 6
**Confidence:** 5

**Summary:**

In “SEED: Towards More Accurate Semantic Evaluation for Visual Brain Decoding” authors proposes a new metric, SEED, to assess the semantic accuracy of images reconstructed from brain activity. They argue that existing metrics (e.g., SSIM, CLIP, Inception) fail to align with human perception of semantic similarity, often overestimating model performance. SEED combines three complementary measures: (1) Object F1, which captures object-level overlap using open-vocabulary grounding models; (2) Cap-Sim, which compares captions of ground-truth and reconstructed images using text embeddings; and (3) EffNet, a feature-based similarity measure. Through medium-scale human evaluations on 1,000 pairs from NSD and additional datasets, SEED metric demonstrates stronger correlation with human judgments than existing metrics and reveals that state-of-the-art decoding models still struggle to reconstruct fine-grained semantics.

**Strengths:**

This is a very timely contribution: Evaluation misalignment between metrics and human perception is a major issue in brain-to-image decoding. In the last 2-3 years I have worked in the field and seen a significant amount of brain decoding works, always competing for few fractions of percentage points of saturated metrics that not always reflect real performance.
Here, SEED addresses a crucial gap that could significantly impact future research.

The metric is well-motivated, interpretable, and partially grounded in both neuroscience and computer vision (object-based attention and semantic binding).

The authors conduct meta-evaluations, robustness tests across models/datasets, and qualitative analyses (worst-case examples, semantic near-miss, and failure modes).

A very good point is the commitment to release the human evaluation dataset adds substantial reproducibility and community value. I'd also suggest the authors to make this metric a very easy to use (ideally a one-liner python function) to encourage reproducibility.

**Weaknesses:**

Introducing the metric is an important point, but I felt something is missing in this work:

Some literature is absent, the number of work and approaches has become very big. Many are missing. To position in the field and constructively criticize the practices I think extended literature review is needed.

This metric is better aligned with human judgement, but still has some bias from the model and limitations. Could the author discuss more in detail these limitations and future work to be improved? Other metrics, ecc.

Overall, this metric is a linear combination of a bunch of others, all relying on some models. In appreciate the point of view even if the technical contribution is somehow limited.

**Questions:**

1) Will this metric/code be released in a easy to access and no-brainer way to encourage wide usage? I think this is really the keypoint here, otherwise the whole thing loses a lot of traction.

2) In the field everyone is competing to have his own line in bold. Your new metric was evaluated on some famous approach, but the best-in class and the order didn't change much. Since the ranking was mostly unaffected, could you highlight better the cases where this metric is telling something more? I liked the failure cases ecc, but I really would love to see more example and use cases where errors or successes were hided by other metrics and SEED is able to point them out

3) The sample size for human evaluation is kind of limited but I understand this is difficult to solve. There is a lot of work on THINGs dataset with million of annotations and human preferences between categories and images ecc, as well as many models trained on them, Does it make sense to compare SEED with these models?

4) One thing that could blow my mind is whether you think SEED could became an objective loss function to be optimized. Right now, as far as I understood it involves several non-differentiable operation that limit the use as a metric alone, but do you think this could be extended as a training objective?

5) Discussion for a position paper like this is a bit short and dry. What are the implication? What's the significance? Why it's needed? Plus general reflection on the field would improve the maturity of the work.

---

> ### Author Response · Authors · 2025-11-20
> **Rebuttal to Reviewer cg7v**
>
> **Weakness 1**
> We included a more detailed introduction of recent literature in Sec. 2. Please take a look!
>
> **Weakness 2**
> Other reviewers also had similar concerns, and we issued a general response regarding comments on the limitations of SEED. Please take a look!
>
> Besides the response, we have added or updated Sec. 6 and D.2 of our manuscript to further discuss the limitations and future directions of SEED.
>
> **Question 1**
> Yes, we also indeed believe this to be an important point. Alongside the publication of our paper, we will fully release all codes and model weights used for our metric on GitHub.
> As the reviewer pointed out, we will make sure that the codebase is well-organized so that users can easily obtain evaluation results by simply providing a (GT, Recon) pair, with a simple pip package installation.
>
> **Question 2**
> As the reviewer pointed out, our proposed metric does not significantly change the ranking of decoding models, and we put a bigger focus on the interpretability of the evaluation results to help researchers discover faults in their decoding model (Sec. 5.4). We took up the reviewer's suggestion and extended the analysis of worst-case judgments (Sec. 5.3) to the existing evaluation metrics, as detailed in Sec. D.1.
>
> To summarize the content, we were able to identify patterns where existing metrics make a misjudgment, while SEED made a human-adjacent judgment. Low-level metrics can give high scores when less relevant pixel structure matches but semantics are wrong (e.g., wrong object), and high-level metrics can overvalue images sharing only coarse features or ImageNet classes. In these cases, SEED typically produces rankings closer to human intuition, showing it can correct many of the failure modes of prior metrics.
>
> **Question 3**
> Following the reviewer’s suggestion, we conducted a literature review to identify an appropriate model trained on the THINGS dataset for evaluating semantic similarity between two images. Specifically, we searched for a model trained on odd-one-out data from the THINGS dataset, which includes human annotations indicating which image deviates from the other two. We were able to locate one such model, gLocal [1].
>
> To be specific, gLocal finds a linear transformation weight based on the THINGS dataset for the transformation of image embeddings from existing pre-trained image encoders to better reflect the human similarity judgments.
>
> We used the **DINOv2-vit-large-p14** and **OpenCLIP-ViT-L/14(LAION-2B)** models to obtain image embedding for GT and its reconstructed images, and measured the similarity based on their cosine-similarity of the two embeddings.
>
> The meta-evaluation results on NSD are as follows.
>
> | Metric           | Pairwise Acc. | Kendall | Pearson |
> |------------------|----------------|---------|---------|
> | gLocal-DINOv2    | 75.1%          | 0.503   | 0.677   |
> | gLocal-OpenCLIP  | 77.7%          | 0.554   | 0.748   |
> | SEED             | **81.0%**      | **0.621** | **0.813** |
>
> Importantly, we emphasize that the interpretability and diagnostic utility of SEED extend beyond its meta-evaluation performance. A key contribution of SEED is not merely providing a single similarity score, but offering interpretable and diagnosable components, Object-F1 for object-level correctness and Cap-Sim for semantic fidelity. While a THINGS-trained models may provide a reasonably high-quality scalar similarity score, it ultimately functions as a black-box measure. In contrast, SEED enables researchers to explicitly identify why a reconstruction failed, as demonstrated in our Sec. 5.4.
>
> [1] : Muttenthaler, Lukas, et al. "Improving neural network representations using human similarity judgments." Advances in neural information processing systems 36 (2023): 50978-51007.

---

> > ### Comment · Reviewer_cg7v · 2025-11-20
> > **Thanks for the rebuttal**
> >
> > I thank the authors for their replies to my comments. Overall I think this is mainly a timely work now that the research on visual brain decoding is increasing and I find this study and the new metric proposed well motivated. The angle taken by authors is interesting and could foster more discussion on how we should measure performance of our models.
> >
> > So, many thanks to the authors for addressing my comments, especially in the short window of time of this rebuttal. I appreciated the comparison with glocal and the revised version of the paper.
> >
> > I'll confirm my scoring. Even though I could support acceptance of this work and I think ICLR could be a good venue for interesting discussion. However some limitations are still here (for example the fact that this metric a combination of other scores, that requires a bunch of models to be computed and this could be both unstable and hard to implement, or the fact that overall this is not extremely more informative than previous ones combined).
> >
> > The most important thing to me is making really straightforward the use of this new metric to encourage usage. As a researcher in the field, I could consider to include it in future evaluations of visual brain decoding works.
> >
> > Wish you good luck!

---

> > > ### Author Response · Authors · 2025-11-23
> > > **Thank you for the feedback**
> > >
> > > First of all, we sincerely thank the reviewer for engaging on a constructive review process, carefully reading our rebuttal, and appreciating the contributions of our work. The comments and the questions by the reviewer have genuinely helped us improve our paper.
> > >
> > > We would also like to share our opinion regarding the limitations the reviewer mentioned. While some instability may arise from relying on existing off-the-shelf models, SEED was designed to mitigate model-specific instabilities in a complementary manner, supported by our analysis in Sec. 5.3.
> > >
> > > Also since we are trying to introduce multiple new metrics, we understand the process may look daunting, but we will do our best to make sure that implementing our evaluation framework would not be difficult at all. We plan to fully open-source our code and provide a pip-installable package to allow users to run the entire evaluation pipeline with only a few lines of code.
> > >
> > > Finally, we emphasize that while SEED leverages off-the-shelf models for its computation, we newly developed two of its components, Object-F1 and Cap-Sim, to better reflect how humans assess image similarity between images based on the models. This allowed SEED to yield a statistically significant improvement over all of the existing metrics in terms of human alignment, as stated in Sec. 5.1.
> > >
> > > More importantly, these metrics were designed to provide better interpretability compared to the existing eight metrics, as their simple and abstract evaluation process made analyzing model faults very difficult.
> > >
> > > As discussed in Sec. 5.4 as well as question 2, this choice allowed us to discover some common failure modes in current visual brain decoding models. We expect researchers would be able to use this information, or discover new failure modes with our metric, to push brain decoding technology even further.
> > >
> > > We would be grateful if the reviewer would reconsider these aspects, should the reviewer find it appropriate.
> > > Once again, we sincerely thank the reviewer for their time and attention invested to reviewing our work and providing constructive feedback.

---

> ### Author Response · Authors · 2025-11-20
> **Rebuttal to Reviewer cg7v (cont'd)**
>
> **Question 4**
> Yes, we have indeed been considering this direction of incorporating SEED into the loss function to further improve existing decoding models. As the reviewer pointed out, however, the current SEED pipeline includes several non-differentiable operations, which prevents its direct use during training. Developing differentiable relaxations of these components so that SEED can be integrated into the objective function represents a promising direction for future work.
>
> However, we also think such training objectives needs to be implemented alongside innovations that lessens the burden of training for brain decoding models. Most of the recent brain decoding models rely on a diffusion module at the end to generate the reconstruction; therefore, if we wish to train the model with a loss that is applied to the final image output, it would incur a substantial memory and computation cost, as it needs to be backpropagated through the whole diffusion process. We believe this is partially why all of the decoding models we tested in Sec. D.3 (MindEye2, NeuroPictor, MindBridge, UniBrain, and BrainGuard) refrain from optimizing the model with the final reconstruction image, and instead opt to train with some intermediate proxy.
>
> **Question 5**
> We agree with the reviewer that our paper needs more discussion, and have added corresponding discussions for Sec. 6, D.1, and D.2. Please take a look!

---

### Official Review · Reviewer_hyma · 2025-10-25

**Soundness:** 3
**Presentation:** 3
**Contribution:** 2
**Rating:** 4
**Confidence:** 4

**Summary:**

This paper proposes SEED (Semantic Evaluation for Visual Brain Decoding) — a new metric designed to better evaluate visual brain decoding models in terms of semantic similarity. The authors integrate three components — Object F1, Cap-Sim, and EffNet — to capture complementary aspects of human-like visual perception. Extensive human evaluations demonstrate that SEED aligns more closely with human judgments than existing metrics.

**Strengths:**

* A more human-like evaluation framework is essential for the brain decoding community.

* The use of large-scale human evaluations is impressive.

**Weaknesses:**

* A key limitation of SEED lies in its reliance on off-the-shelf captioning and detection models (e.g., GIT, Grounding-DINO). These components were not trained to reflect human semantic judgments but to optimize task-specific objectives (caption likelihood or object detection accuracy). As a result, SEED may inherit their systematic errors, leading to misleading evaluations in certain cases.

* The authors argue that in some existing metrics such as n-way identification, decoding models have reached near-ceiling performance, but this is largely due to the small candidate pool typically used in evaluation. If the number of candidate images is substantially increased (e.g., 100-way or 1,000-way identification), most current decoding models exhibit significant performance degradation, revealing substantial room for improvement.

* Moreover, recent progress in visual decoding is moving toward multimodal decoding frameworks (text and image). In such settings, the semantic quality of reconstructed images can already be assessed through predicted text. This trend raises questions about the necessity of using GIT-based caption generation in SEED. Since GIT introduces additional linguistic biases and noise unrelated to the decoding model itself.

* The complexity of SEED may to some extent hinder its widespread adoption. For example, its reliance on multiple models introduces issues such as version differences, model updates, and parameter inconsistencies, which reduce the reproducibility of results and the comparability within the community.

Minor: Several recent works such as [1-2] in brain decoding are not cited or discussed.

[1] Bridging the Gap between Brain and Machine in Interpreting Visual Semantics: Towards Self-adaptive Brain-to-Text Decoding, ICCV 2025.

[2] Mindgpt: Interpreting what you see with non-invasive brain recordings, IEEE TIP 2025.

**Questions:**

* Could SEED be adapted to multimodal decoding (e.g., brain-to-text or cross-modal tasks)?

* Could evaluating the semantic fidelity of reconstructions using fine-grained category labels rather than object detection models provide a more efficient and reliable assessment?

---

> ### Author Response · Authors · 2025-11-20
> **Rebuttal to Reviewer hyma**
>
> **Weakness 1**
> Please refer to general response regarding comments on the limitations of SEED!
>
> In order to minimize the bias introduced, the evaluation process that we propose was inspired by the human visual perception pipeline, and we empirically demonstrated that our proposed metrics are the most human aligned.
>
> In Sec. 5.2, we also conducted an experiment to test the robustness of SEED by interchanging the different components, to see if SEED is by the specific captioning or detection model, and we discovered the choice of model did not really affect the performance of SEED.
>
> **Weakness 2**
> As the reviewer pointed out, the near-ceiling performance observed under popular evaluation practices (i.e., 2-way identification) could be alleviated by increasing the size of the candidate pool. However, we emphasize that $n$-way identification itself has fundamental limitations, as discussed in Sec. 3.2.
>
> First, comparing 2-way identification scores across different models is inherently inappropriate. In this setting, each reconstruction is compared only against other reconstructions produced by the same decoding model, meaning that the candidate pools differ across models. Consequently, each model is evaluated under different comparison criteria, making direct comparison of $n$-way identification scores invalid.
>
> Second, although one could artificially “lower” performance by increasing $n$, computing $n$-way identification requires evaluating $|\mathcal{T}|^n$ comparisons ($|\mathcal{T}|$ being the size of the test set), which quickly becomes intractable as $n$ grows.
> However, we were able to identify the following workaround: if $p$ denotes the 2-way identification accuracy for a single (GT, Recon) pair, the corresponding $n$-way accuracy would be $\frac{{|\mathcal{T}|p -1} \choose {n-1}}{{|\mathcal{T}|-1} \choose {n-1}}$, which is a function of $p$. Nevertheless, this does not provide any meaningful additional insight into model performance, as the n-way identification accuracy only widens the gap between 100\% and the 2-way identification accuracy for each pair by a set amount.

---

> ### Author Response · Authors · 2025-11-20
> **Rebuttal to Reviewer hyma (cont'd)**
>
> **Weakness 3**
> As the reviewer pointed out, recent progress has been made in multimodal brain decoding, where models simultaneously output text and visual information from fMRI. However, we would like to clarify that the evaluation of reconstructions for each modality is typically performed separately.
>
> For example, [1, 2, 3] evaluates text reconstructions using METEOR, ROUGE, and CLIP text encoder–based semantic metrics, while visual reconstructions are assessed using eight existing metrics.
>
> We further note that [4, 5] mentioned later by the reviewer primarily focus on brain-to-text decoding; their models do not reconstruct the visual stimuli themselves during inference.
> In particular, [5] generates images by feeding the decoded text into Stable Diffusion, which leads to substantial loss of visual detail due to the two-stage reconstruction pipeline. Therefore, we believe that [5] does not fall into the category of multimodal brain decoding models in the same sense as [1,2,3].
>
> Although [5] employs CLIP text embedding based-evaluation to assess visual reconstruction quality, by measuring the cosine similiarity between text embeddings of GT image captions and their corresponding image embeddings generated from the brain-decoded texts, this approach is not widely adopted for evaluating visual outputs in the visual brain decoding literature.
> This scheme resembles evaluation practices in the text-to-image (T2I) generation field; however, as we clarified in our response to Reviewer UN34 (Q1), evaluation in T2I generation and visual brain decoding differ fundamentally in their goals. We kindly ask the reviewer to refer to that response for additional context.
>
> As the reviewer suggested, if a visual decoding model also produces text reconstructions, these can indeed be incorporated into Cap-Sim. However, it is important to note that SEED is designed specifically for evaluating visual reconstructions from any visual brain decoding model. That is, the metric should be able to measure similarity based solely on the ground-truth and reconstructed images. Since the majority of current decoding models focus on generating images only, captioning models are necessary for text-based evaluation. Moreover, ground-truth captions are not always available, for instance, the GOD dataset contains images without corresponding captions. Therefore, generating captions with a captioning model is required for this evaluation.
>
> We also agree with the reviewer that biases and noise can be introduced by the captioning model, as discussed in Sec. 5.3. Nevertheless, we believe that the combination of the three metrics in SEED effectively complements each metric’s individual limitations, resulting in a more robust evaluation framework.
>
> [1] Xia, Weihao, et al. "Umbrae: Unified multimodal brain decoding." European Conference on Computer Vision. Cham: Springer Nature Switzerland, 2024.
>
> [2] Shen, Guobin, et al. "Neuro-vision to language: Enhancing brain recording-based visual reconstruction and language interaction." Advances in Neural Information Processing Systems 37 (2024): 98083-98110.
>
> [3] Mai, Weijian, and Zhijun Zhang. "Unibrain: Unify image reconstruction and captioning all in one diffusion model from human brain activity." arXiv preprint arXiv:2308.07428 (2023).
>
> [4] Chen, Jiaxuan, et al. "Bridging the Gap between Brain and Machine in Interpreting Visual Semantics: Towards Self-adaptive Brain-to-Text Decoding." Proceedings of the IEEE/CVF International Conference on Computer Vision. 2025.
>
> [5] Chen, Jiaxuan, et al. "Mindgpt: Interpreting what you see with non-invasive brain recordings." IEEE Transactions on Image Processing (2025).
>
> **Weakness 4**
> We will later share the current version of code and weights for each components through GitHub to eliminate the inconsistencies.
>
> We also want to mention that this is a problem not unique to SEED, but for any metric that decides to use AI-supported evaluation. We will therefore do our best to help other researchers replicate our results by sharing the model weights and code, and even if the version differs, as long as people use the same version of the evaluation model to evaluate decoding models, the comparison will be fair.
>
> **Weakness 5 (Minor)**
> We included these works in the Sec. 2. Please take a look!

---

> ### Author Response · Authors · 2025-11-20
> **Rebuttal to Reviewer hyma (cont'd)**
>
> **Question 1**
> As long as cross-modal decoding models produce image reconstructions as outputs, our framework can be applied to evaluate those images. Since SEED is specifically designed to evaluate how closely reconstructed images match the corresponding ground truth image, it is not directly applicable to brain-to-text decoding tasks.
>
> As a side note, we believe that evaluating text reconstructions is comparatively less subtle and challenging. Text outputs can be assessed using exact-match–based metrics, and their semantic content can be evaluated with pre-trained text encoders, in a manner similar to Cap-Sim.
>
> **Question 2**
> To be honest, we could not fully understand what "using fine-grained category labels rather than object detection models" exactly mean, and we would be happy to provide a detailed response once further clarification is given. We will provide an answer based on our current understanding of the question. Using more fine-grained category labels compared to what we currently use may be a possible direction. However, for such an evaluation protocol, we would need to extract information about which objects are present in both the ground-truth images and their reconstructions. This would require incorporating object detection or tagging models into the evaluation pipeline to pick out which specific category is needed.
>
> In the past we tried experimenting with high-performance tagging models like RAM++ [1], and we found the automatically generated tags to be quite noisy, especially for the distorted reconstructions. The tagging model also tended to generate multiple synonyms for a same concept: for example, for a surfing image, the model might generate surf, surfing, surfboard, surfer, surfing man, etc. These noisy and duplicate categories made it difficult for us to establish a clean process to calculate metrics such as Object F1, and we found that our current formulation of object categories works better in practice.
>
> While it would be possible to manually annotate fine-grained category labels for evaluation, we believe the reviewer would agree that the associated human labor cost would be prohibitively high.
>
> [1] Huang, Xinyu, et al. "Open-set image tagging with multi-grained text supervision." Proceedings of the 33rd ACM International Conference on Multimedia. 2025.

---

> ### Author Response · Authors · 2025-11-28
> **A gentle reminder for reviewer hyma**
>
> Thank you again for your time and effort in reviewing our paper.
>
> We wanted to kindly remind you that less than a week remains for the discussion period, and we would be grateful for any additional comments or concerns you may have before the discussion phase draws to an end.
>
> Thank you for your efforts!
>
> Authors

---

### Official Review · Reviewer_UN34 · 2025-10-30

**Soundness:** 3
**Presentation:** 3
**Contribution:** 3
**Rating:** 6
**Confidence:** 3

**Summary:**

This paper identifies a critical problem: current evaluation metrics are poorly aligned with human judgment of semantic similarity in visual brain information reconstruction. Even if results achieve near-perfect scores on existing metrics, they may still be semantically flawed. To address this, they propose SEED, a new evaluation metric that integrates three complementary components—Object F1, Cap-Sim, and EffNet. Through extensive "meta-evaluation" against a new human judgment dataset, they demonstrate that SEED aligns significantly better with human evaluation than all existing metrics.

**Strengths:**

1. The motivation of the article is very good. The current evaluation of decoding models indeed has such a problem. Especially, many tasks utilize contrastive learning in the CLIP space and then proceed with image generation. Naturally, this leads to excellent generation scores.

2. The design of SEED is thoughtful. The two new proposed metrics, Object F1 (object-level attention) and Cap-Sim (feature binding into a scene description), offer novel and complementary perspectives. It is indeed something worth noting during the recovery process.

3. The article conducted numerous human alignment experiments and evaluated the results of many models, proving that the proposed indicators are indeed closer to human understanding of image restoration.

**Weaknesses:**

1. The proposed indicators mainly focus on assessing semantics and may be insensitive to information like color and texture. This is also a significant factor influencing restoration and human perception.

2. The article does not provide a clear description of how human evaluations are conducted. For example, in Figure 1, why is that image ranked 846th out of 1000? How was this ranking determined? The human evaluators rated both "semantic and perceptual similarity". For the meta-evaluation, which rating was used?

3. The method used in the article is reasonable but somewhat complex, which is composed of many current models. I'm not sure if there might be any hidden biases here. If one of the models has a problem, will it lead to the failure of the evaluation? This merely presents a feasible solution, but does not discuss whether there are better evaluation methods or directions for improvement.

**Questions:**

1. In terms of semantic understanding, this is a problem that all generative models encounter. Does this method and insight also work for current generative models? Or is there any new semantic metric in the field of image generation now? How are they being used?

2. Given that multiple large models were used, what was the speed of the inference? What is the total amount of computing resources consumed for the operation? If it is extremely large, it may affect the use of the indicators. Besides, how and why are the weights of multiple models determined?

---

> ### Author Response · Authors · 2025-11-20
> **Rebuttal to Reviewer UN34**
>
> **Weakness 1**
> We recognize that SEED may capture color and texture information less explicitly than SSIM or existing low-level evaluation metrics. Nonetheless, SEED remains sensitive to these low-level aspects, which still affect its evaluations.
> Notably, $\overline{\text{EffNet}}$, one of the components of SEED, compares features from a pre-trained EfficientNet model. Because EfficientNet is originally trained for ImageNet classification, it must learn to distinguish objects using fine-grained visual cues. For instance, separating different species of birds, flowers, or animals requires encoding rich texture and color information. Consequently, the similarity between EfficientNet features of the ground-truth and reconstructed images inherently reflects differences in color and texture as well.
>
> **Weakness 2**
> Please note that the details on how the human evaluations were conducted are provided in Sec. A of the supplementary material.
>
> The rank presented in Fig. 1 as well as in Sec. 5.3 corresponds to the rank of the averaged semantic similarity of (GT, Recon) pair obtained from human evaluations of a total 1,000 pairs. Each evaluator’s semantic similarity scores are z-normalized along the evaluator’s own scoring axis to account for individual differences in scoring variance, and subject-wise averaged scores were used to rank each pair.
>
> We understand our explanation of the rankings was insufficient, and we updated our manuscript Sec. 5.3 to further explain how the rankings were obtained.
>
> For the meta evaluation throughout our study, we used the human ratings of semantic similarity. We did collect perceptual similarity alongside the semantic similarity, but we decided to not use it for this study since we wanted to focus on the semantic similarity. We will release all of our human scores so that future researchers can use the perceptual parts.
>
> **Weakness 3**
> Since SEED is a combination of three different metrics, as for cases like the ones presented in Sec. 5.3, we found SEED to make reasonable judgments when only one of the three metrics make a misjudgment. We also newly added Sec. D.2 to show cases where all metrics fail to make a human-aligned judgment when an unusual or malformed image is given as the reconstruction, which leads to the failure of SEED, and discussed directions of improvement for SEED and future metrics.
> Also please take a look at our general response regarding the limitations of SEED!

---

> ### Author Response · Authors · 2025-11-20
> **Rebuttal to Reviewer UN34 (cont'd)**
>
> **Question 1**
> In the text-to-image generation (T2I) field, several evaluation metrics have recently been proposed.
>
> To assess how well generated images follow user prompts, CLIP-FlanT5 [1] leverages an in-house visual question answering (VQA) model to evaluate prompt adherence.
> In addition, a number of metrics target the aesthetic and overall quality of generated images. ImageReward [2] collects large-scale human evaluation data and trains an evaluation model that estimates human preference for each (text, image) pair. Similarly, MPS [3] gathers human preference data and trains an evaluation model that captures a broader range of criteria, including aesthetics, alignment, and overall image quality.
>
> However, we emphasize that the evaluation setup in visual brain decoding is fundamentally different from that of T2I.
> The T2I setting primarily assesses how well generated images align with the input prompts, and crucially, no ground-truth image exists. In contrast, visual brain decoding provides a ground-truth (GT) image, and evaluation focuses on how closely the reconstruction matches the GT image.
>
> We also emphasize that the task we are dealing with is unique to brain decoding: we are comparing the similarity of two images which can be very different. In that sense, image compression is one of the most similar tasks since it compares two images. However, since compressed images are not to different from the original, people focus more on perceptual fidelity instead of semantic comparisons, because compressed images are not different to the degree that the semantics change.
>
> Due to there being no major fields that require the direct similarity comparison of two different images, evaluation methods for brain decoding models are quite unique to this field. We believe this is why metrics dedicated to visual brain decoding are still underdeveloped, and a study dedicated to how brain decoding models are evaluated is imperative at this point.
>
> Moreover, we experimentally found that directly applying existing T2I evaluation models to visual brain decoding yields poor alignment with human judgments. Specifically, we generated captions for the GT images and evaluated prompt–reconstruction alignment using CLIP-FlanT5. As shown in the 7th–8th rows of Tab. 5 in Sec. C.2, these meta-evaluation results underperform compared to SEED.
>
> [1] : Lin, Zhiqiu, et al. "Evaluating text-to-visual generation with image-to-text generation." European Conference on Computer Vision. Cham: Springer Nature Switzerland, 2024.
>
> [2] : Xu, Jiazheng, et al. "Imagereward: Learning and evaluating human preferences for text-to-image generation." Advances in Neural Information Processing Systems 36 (2023): 15903-15935.
>
> [3] : Zhang, Sixian, et al. "Learning multi-dimensional human preference for text-to-image generation." Proceedings of the IEEE/CVF Conference on Computer Vision and Pattern Recognition. 2024.
>
> **Question 2**
> We note that since we only have to do inference, the cost of our proposed evaluation methods is fairly cheap. The cost of running and training the actual brain decoding models is usually much higher than the evaluation process.
> We measured the VRAM usage and computational time for each component of SEED with a single A6000 GPU.
> Both are measured with a batch size of 1 on 1,000 images or (GT, Recon) pairs, and there will be a trade-off between the VRAM usage and the computation time if the batch size is adjusted.
>
> The results are summarized as below.
>
> | Metric               | VRAM Usage (GB) | Computation Time (s) |
> |----------------------|-----------------|-----------------------|
> | Object F1            | 4.46            | 528.42                |
> | GIT Captioning       | 3.25            | 268.11                |
> | Sentence Transformer | 0.53            | 5.39                  |
>
> For the final question regarding "weights of multiple models," we were unsure if this meant the weight for each model that was used to calculate SEED in Equation (7), or the model weights (or model versions) of each model. We will post the answer for both questions.
>
> For Equation (7), we decided to mix the three metrics with a simple 1:1:1 ratio to avoid introducing any biases that might arise when we decide to optimize the mixing ratio of the three metrics. It would be possible to optimize the weights based on the human survey results, but doing so has the risk of introducing biases based on our survey results, and without another external human survey that we could use as a reference, we did not have a reliable way to confirm nor deny the existence of such bias.
>
> The model weights (or model versions) were selected considering multiple factors, including their performance on popular benchmarks, generalization capability, ease of access, and computational efficiency.

---

> ### Author Response · Authors · 2025-11-28
> **A gentle reminder for reviewer UN34**
>
> Thank you again for your time and effort in reviewing our paper.
>
> We wanted to kindly remind you that less than a week remains for the discussion period, and we would be grateful for any additional comments or concerns you may have before the discussion phase draws to an end.
>
> Thank you for your efforts!
>
> Authors

---

### Author Response · Authors · 2025-11-20
**General Response**

First of all, we deeply appreciate the valuable criticism and insights from the three reviewers. We are happy to improve our work based on the reviewers’ suggestions.

We also thank the reviewers for acknowledging the strengths and significance of our contribution. As noted, proposing a new evaluation metric for visual brain decoding is both timely and well-motivated. Our metric is grounded in neuroscientific considerations and demonstrates robust alignment with human judgments across different models and datasets. Furthermore, we provide detailed failure-mode analyses of current decoding models and release our human-evaluation dataset to support future research in this area.

At the same time, we take seriously the constructive concerns raised in the reviews. These include the potential failures introduced by relying on existing models, concerns regarding reproducibility and usability of the open-sourced evaluation pipeline, and several additional points raised by individual reviewers.

We have done our best to understand and address each reviewer’s feedback. We also recognize that some concerns may not be fully resolved within the scope of the rebuttal. If that is the case, we would be more than happy to clarify further during the discussion phase.

**The feedback from reviewers and our response to the questions are reflected in the manuscript or the appendix, with additional experiments, figures, and discussions. Any new changes are written in blue text, so feel free to check it out if you are interested.** Any new feedback or questions regarding these changes are also welcome!

Once again, we sincerely thank the reviewers for their thoughtful comments and look forward to a constructive and engaging discussion to further improve our work.

### **Regarding Comments on the Limitations of SEED.**
As all three reviewers noted, SEED relies on off-the-shelf models, which means it may inherit their biases or errors. Consequently, there can be cases where SEED’s semantic similarity assessments deviate from human evaluation.

We agree with the reviewers' sentiment, however we want to stress it is not a problem unique to SEED as most existing evaluation metrics (AlexNet, Inception, CLIP, EffNet, SwAV) also rely on AI models to extract image features.

As discussed in Sec. 5.3, each component of SEED may demonstrate undesirable biases for some specific cases. As shown in those examples, we believe that these limitations can be partially diluted by combining the three metrics, complementing each other.

Nevertheless, we admit that there can be cases where SEED still deviates from human evaluation. We showcased these failure cases in the revised manuscript, and also added a discussion about this in the Sec. D.2, while discussing a possible direction for advancing the evaluation metric.

---

### Author Response · Authors · 2025-12-02
**Summary of the paper, reviews, and discussion.**

### **Paper summary.**

We propose SEED, a semantic evaluation framework for visual brain decoding models. We created two new metrics, Object F1 (object-presence based) and Cap-Sim (caption-embedding similarity), and combined them with a commonly used metric, EffNet, to compose SEED. SEED is meta-evaluated against our new 1,000-pair human judgment dataset and shows higher agreement with human semantic similarity ratings than all eight commonly used metrics. Due to its human-likeness, SEED unlocks the opportunity to analyze state-of-the-art decoding models, revealing systematic semantic errors such as “semantic near-misses” and reconstructions that capture objects but miss crucial contextual details.

### **Strengths.**

Reviewers mostly agree that (1) addressing semantic alignment with human judgments is critical in a field where current metrics are near-saturated yet reconstructions remain semantically flawed; (2) the newly proposed metrics (Object F1, Cap-Sim, and SEED) are well-motivated, and the decomposition of SEED into object-level (Object F1), caption-level (Cap-Sim), and feature-level (EffNet) scores is conceptually appealing and interpretable; and (3) the human-alignment experiments and robustness checks across datasets/models credibly support the main claims.


### **Key concerns and corresponding rebuttals.**

Please refer to the general response for common concerns from all three reviewers and our corresponding response!

Q. Does this apply to image generation models, or is there a new metric for generative models? (UN34 Question 1)

A. The task we are aiming to evaluate (visual brain decoding) is unique in the sense that there is a target ground-truth image. This requires the comparison of the similarity of two images, and with those two images potentially being very different, makes this task fairly unique to visual brain decoding.

Q. Can the semantics be evaluated with the predicted text of multimodal decoding models? Why do we have to use GIT-generated captions? (hyma Weakness 3)

A. SEED is designed to evaluate any visual brain decoding model, with or without separate text predictions. That is, the metric should be able to measure similarity based solely on the ground-truth and reconstructed images. Since the majority of current decoding models focus on generating images only, captioning models are necessary for text-based evaluation. Moreover, ground-truth captions are not always available: for instance, the GOD dataset contains images without corresponding captions.

Q. Could you highlight the cases where the proposed metric is telling something more, for example cases where SEED reveals new strengths and weaknesses of decoding models? (cg7v Question 2)

A. As suggested, we put a big focus on the interpretability of the evaluation results to help researchers discover faults in their decoding model. To summarize our new analysis, we were able to identify patterns where existing metrics make a misjudgment, while SEED makes a human-adjacent judgment. Low-level metrics can give high scores when less relevant pixel structure matches but semantics are wrong (e.g., wrong object), and high-level metrics can overvalue images sharing only coarse features or ImageNet classes. In these cases, SEED typically produces rankings closer to human intuition, showing it can correct many of the failure modes of prior metrics.

---

### Meta-Review · Area_Chair_2d6Q · 2026-01-12

**Summary:**

This paper proposes SEED, a composite metric for evaluating semantic similarity in visual brain decoding, and introduces two new components (Object F1 and Cap-Sim) alongside an EfficientNet feature metric. All three reviewers agree the problem is timely and important, given that existing metrics are near-saturated yet misaligned with human semantic judgment. Strengths consistently highlighted include the strong motivation, the use of a new human evaluation dataset, extensive meta-evaluation across datasets and models, and the interpretability of SEED’s components.

The main concerns raised were: (i) reliance on off-the-shelf models and the potential biases and instability this introduces; (ii) the complexity and reproducibility/usability of the proposed evaluation pipeline; (iii) whether SEED offers fundamentally new insight beyond combining existing metrics; and (iv) clarity and depth of discussion, positioning, and limitations. Reviewers also noted missing literature and initially insufficient explanation of the human evaluation protocol.

Overall, reviewers found the work solid, well-motivated, and potentially impactful for the community, though with some remaining reservations about methodological dependence on pretrained models and the incremental nature of the technical contribution. After rebuttal, most concerns were partially addressed, and the consensus leans positive, with two reviewers above the acceptance threshold and one slightly below but open to acceptance.

**Reviewer Concerns:**

Most of the concerns are well addressed.

**Reviewer Scores:**

Likely to be all postive.

---

### Decision · Program_Chairs · 2026-01-26

Accept (Poster)